# Revealing Impacts of Human Activities and Natural Factors on Dynamic Changes of Relationships among Ecosystem Services: A Case Study in the Huang-Huai-Hai Plain, China

**DOI:** 10.3390/ijerph191610230

**Published:** 2022-08-17

**Authors:** Longyun Deng, Yi Li, Zhi Cao, Ruifang Hao, Zheye Wang, Junxiao Zou, Quanyuan Wu, Jianmin Qiao

**Affiliations:** 1College of Geography and Environment, Shandong Normal University, Jinan 250358, China; 2Shandong Institute of Eco-Environmental Planning, Jinan 250101, China; 3Institute of Geographic Sciences and Natural Resources Research, Chinese Academy of Sciences, Beijing 100101, China; 4School of Soil and Water Conservation, Beijing Forestry University, Beijing 100083, China; 5Kinder Institute for Urban Research, Rice University, Houston, TX 77005, USA

**Keywords:** constraint line, characteristic value, threshold, landscape index, interaction effect

## Abstract

Understanding the dynamic changes of relationships between ecosystem services (ESs) and their dominant factors can effectively adjust human activities to adapt proactively to global climate change. In this study, the Huang-Huai-Hai Plain (HHHP) was selected to assess the dynamics of four key ESs (NPP, net primary productivity; WY, water yield; SC, soil conservation; FP, food production) from 2000 to 2020. The constraint lines of interactions among ESs were extracted based on a segmented quantile regression model. On this basis, the effects of both human activities and natural factors on the key features of the interactions between ESs were quantified with the help of automatic linear model. The results indicated that two types of constraint relationships, including exponential and humped-shaped, existed among the six pairs of ESs. In the past two decades, small changes in NPP thresholds would lead to large variations in other ESs thresholds. Precipitation and normalized difference vegetation index were the key factors to determine the constraint strength of ESs in the HHHP. The potential maximum value of WY in the HHHP could be increased by adjusting landscape shape to make it more complicated. This study helps to improve the potential of target ESs and provides a decision-making basis for promoting regional sustainable development.

## 1. Introduction

Ecosystem services (ESs) refers to the benefits that human can directly or indirectly derive from ecological system (ES), mainly including supply services (e.g., food to sustain human survival), support services (such as maintaining water cycle of life on earth’s living environment), regulating services (flood regulation, heat island adjustment, etc.) and cultural services (e.g., recreation), which are critical in landscape optimization and ecosystem management [1,2,3]. Due to diversity, spatial heterogeneity, and human preference for ESs, when humans selectively emphasize one type of ES, it often compromises the provision of one or more other services, leading to an unintended decline in ESs and potentially causing a range of environmental problems [4]. Since the 1970s, with the impact of climate change and human activities on the environment, about 60% of ESs in the world have been declining [5,6]. Degradation of ecosystem functions will directly threaten regional and global ecological security, endangering the well-being of present and future generations [7], and certain ESs management measures are urgently needed.

Scientifically quantifying and clarifying the relationship between ESs has important theoretical support significance for promoting ecosystem sustainability and improving human well-being [4]. ESs relationships mainly includes trade-offs and synergies [8,9,10,11,12]. Previous studies mostly focused on the identification of trade-off/synergistic features, the analysis of spatial and temporal variability, and the exploration of driving factors [13,14]. For example, Gou et al. used correlation analysis and K-means clustering to explore the spatial distribution patterns of trade-offs and synergies of ESs in different service clusters in the Three Gorges reservoir area [10]. Kubiszewski et al. estimated the future value of ESs and possible relationships among services by simulating land use under future scenarios [11]. Raudsepp-Hearne et al. used correlation coefficients to reflect the strength of trade-offs between provisioning and regulating services in the urban landscape of Quebec, Canada [12]. However, when conducting trade-off and synergistic relationship analysis, it is often assumed that the relationship between ESs is monotonous. This analysis method ignores the complexity of the joint influence of multiple factors on ESs, especially when ESs relationship presents a cloud-like distribution pattern. In this case, the ecological mechanisms behind ESs are difficult to explain and express by the trade-off/synergy analysis.

Compared with traditional correlation and regression analysis, the constrained line method proposed by Hao et al. and Qiao et al. can better characterize the interaction relationship between ESs with scattered cloud distribution [15,16]. The constraint line indicates that one service is influenced by the constraint of another service. The points on the constraint line indicate that other factors have little or no influence on the response variable. Although the relationships among ESs are limited by regional characteristics and there is spatial heterogeneity in the types of constraint relationships, the proposed constraint relationship is an effective complement to the relationship between ESs, which is conducive to promoting the realization of regional landscape sustainability.

Quantitative evaluation of the relationship between ESs and optimal management can reasonably plan ecological functional areas and improve ecological protection proposals and is also a necessary step to promote coordinated regional development. It is worth noting that the relationship between ESs is not constant, and human activities and climate change often lead to complex, heterogeneous, and fragmented landscape patterns [17], which in turn affect regional ESs [18,19,20], resulting in temporal variability in the relationship between ESs [21,22]. For example, Lin et al. found that as urbanization continued, the landscape of natural ecosystems gradually fragmented, leading to a decline in ESs [23]. Socioeconomic processes (e.g., demographic and economic changes) can also affect the stability of ESs [24,25]. Zhang et al. revealed a complex relationship between ESs and socioeconomics, showing that increases in population density and gross domestic product (GDP) per capita all have negative effects on ESs [26], altering the interaction relationship between ESs. Qiao et al. further demonstrated that the trade-offs and synergies between ESs change over time and are spatially dependent [27]. Therefore, the effects of factors such as human activities and climate change on ESs relationships should be considered when optimizing ESs relationships. At present, most studies still focus on the analysis of factors influencing ESs, and few studies evaluate the variation in ESs relationships and lack the main causes and mechanistic explanations for the dynamic changes of their relationships. Clarification of the above issues can help provide a scientific basis for regional ecosystem optimization and management decisions.

Huang-Huai-hai Plain (HHHP) is an important food supply area in China, which can provide a variety of ESs such as food supply and soil conservation. However, due to the impact of human activities and climate change, the regional landscape pattern changes dramatically, and the ecological environment is fragile, which easily leads to the loss of ESs [28,29]. How to optimize the regional ecosystem, improve ESs potential and promote regional sustainable development is an important issue in HHHP. To address the above scientific questions, the objectives of this study are as follows: (1) to quantify the constraint relationships among six pairs of ESs from 2000 to 2020; (2) to identify the characteristic values (threshold, slope (k), and constant term (b)) of the constraint interactions among ESs and characterize their changes; (3) to reveal the effects and mechanisms of human activities and natural factors and their interactions on the constraint interactions among ESs to provide insights for ESs management.

## 2. Materials and Methods

### 2.1. Study Area

HHHP is located at 112°43′–112°71′ E, 32°49′–40°57′ N, spanning Beijing and Tianjin cities, and Hebei, Shandong, Jiangsu, Henan, and Anhui provinces, with an area of about 3.9 × 10^5^ km^2^ (Figure 1). It belongs to the mid-latitude monsoon climate, characterized by four distinct seasons and large temperature (TEM) differences. The annual average TEM is about 8–15 °C, and the annual average precipitation (PPT) is about 500–1000 mm. The temporal and spatial distribution of PPT is uneven and the seasonal characteristics are obvious. Summer accounts for about 70% of the annual rainfall [30]. The HHHP is low-lying and flat, and the land use type is mainly cultivated land, which accounts for more than 70% in 2020 (Figure 2), making it the largest agricultural region in China [14,31]. From 2000 to 2020, the GDP of the HHHP increased from 157 million yuan to 1.338 billion yuan, and the construction land area increased by 1.9 × 10^4^ km^2^. Under the influence of human activities, the regional landscape pattern has undergone large changes (Figure 2) and environmental pressure is increasing [31], which seriously threatens ecological security and food supply in China.

### 2.2. Data Sources

The data used in this study mainly include meteorological data, DEM data, soil data, land use/cover, NDVI, crop yield, and socio-economic data. Table 1 gave the datasets used to evaluate the four ESs and their brief descriptions. Land use data were classified according to the first-level classification system, including construction land, forest land, grassland, cultivated land, water area, and unused land. Modis Reprojection Tool was used for batch processing of NDVI remote sensing data. Meteorological station data were interpolated by the kriging method to obtain TEM and PPT in each grid and used for driving ES assessment models. Among them, GDP was only available for four periods 2000, 2005, 2010, and 2015, and the rest of the years were obtained by fitting the statistical data of each city. To ensure uniform data spatial resolution, all data were resampled to 250 m × 250 m. In addition, all spatial figures were drawn with the ArcGIS 10.6 platform.

### 2.3. Modelling the ESs from 2000 to 2020

Based on the regional characteristics of HHHP, the amount of net primary productivity (NPP), food production (FP), soil conservation (SC), and water yield (WY) were selected as the four key ESs in this study. The research framework of this study was shown in Figure 3.

#### 2.3.1. Net Primary Productivity

NPP refers to the organic matter produced by green plants through photosynthesis and is an important component of the ecosystem carbon cycle. The calculation of NPP was mainly based on the Carnegie-Ames-Stanford mode (CASA) [32,33]. The calculation formula is as follows.
(1)$$NPP\left(x,t\right)=APAR\left(x,t\right)\times \varepsilon \left(x,t\right),$$
where $NPP\left(x,t\right)$ is the net primary productivity of vegetation (g C); *x* denotes the grid; *t* denotes the month, including all months throughout the year; $APAR\left(x,t\right)$ is the solar radiation energy effectively absorbed by vegetation during photosynthesis (MJ m^−2^), while $\varepsilon \left(x,t\right)$ is the actual light energy utilization rate of vegetation during photosynthesis (g C MJ^−1^).

#### 2.3.2. Food Production

The calculation of FP was based on the linear relationship that exists between NDVI and grain yield to obtain the raster data of FP [34,35]. The equation is as follows.
(2)$$FP_{i}=\frac{NDVI_{i}}{NDVI_{sum}}\times G_{sum},$$
where $FP_{i}$ is the food production allocated to grid *i* (t ha^−1^); $NDVI_{i}$ is the NDVI of grid *i* in cropland; $NDVI_{sum}$ is the sum of NDVI of cropland in a city; $G_{sum}$ is the total food production in a city (t ha^−1^), and the food types in this study include rice, wheat, maize, beans, and tubers.

#### 2.3.3. Soil Conservation

SC refers to the erosion control capacity of an ecosystem to prevent soil erosion and the ability to store and retain sediment [36]. In nature, excessive soil erosion causes loss of soil fertility, reduced agricultural yields, and degradation of rivers, lakes, and estuaries. This study adopted a Revised Universal Soil Loss Equation (RUSLE) to calculate the soil conservation service, which was based on the difference between the potential erosion of soil without vegetation cover and the erosion of soil with real vegetation cover [10,37]. The equations are as follows.
(3)$$A_{p}=R\times K\times LS,$$
(4)$$A_{r}=R\times K\times LS\times C\times P,$$
(5)$$SC=A_{p}-A_{r},$$
where $A_{p}$ and $A_{r}$ are potential soil erosion and actual soil erosion (t hm^−2^), respectively; *R* is rainfall erosion force factor (MJ mm hm^−2^ h^−1^); *K* is soil erodibility factor (t h MJ^−1^ mm^−1^); *LS* is topography factor, where *L* represents slope length and *S* represents slope; *C* is vegetation cover factor; and *P* is soil and water conservation factor. The details of parameters are described in the Appendix A.

#### 2.3.4. Water Yield

WY refers to the water supply part of PPT excluding transpiration, which is evaluated by the Integrated Valuation of Ecosystem Services and Tradeoffs (InVEST) model. The water production module of InVEST model is based on the principle of water balance and consists of three parts: soil moisture, surface runoff and water trapped by litters and vegetation canopy [37,38]. The calculation formulas of WY are as follows:(6)$$WY_{x}=\left(1-\frac{EET_{x}}{PPT_{x}}\right)PPT_{x},$$
(7)EETxPPTx=1+PETxPPTx−⌊1+PETxPPTxW⌋1W,
(8)$$PET_{x}=Kc_{x}\times ETo_{x},$$
(9)$$W_{x}=\frac{AWC_{X}\times Z}{PPT_{x}}+1.25,$$
where $WY_{x}$ is the annual water yield of grid *x* (mm); $EET_{x}$ is the annual actual evapotranspiration of grid *x* (mm); $PPT_{x}$ is the annual PPT of grid *x* (mm); $PET_{x}$ is the annual average potential evapotranspiration of grid *x*; $Kc_{x}$ represents the vegetation evapotranspiration coefficient; $ETo_{x}$ is the reference vegetation evapotranspiration of grid *x*; $AWC_{X}$ indicates the available water of vegetation (mm); $W_{x}$ is a non-physical parameter; *Z* is a seasonal constant. The details of parameters are described in the Supplementary Material.

### 2.4. Extraction of the Constraint Lines between Paired ESs

The constraint line can describe the constraining effect of constraint variables on response variables in complex ecosystems affected by multiple factors [39,40]. It was first proposed by Scharf et al. [41] and later applied to the evaluation of ESs relationships [16]. Points on the constraint line indicate that the other variables have the least effect on the response variable. At present, there are four main drawing methods for constraint lines: parameter method, scatter cloud grid method, quantile regression method, and quantile segmentation method [15]. In this study, quantile segmentation was used to extract the constraint relationship types between six pairs of ESs (NPP_SC, NPP_WY, FP_NPP, FP_WY, FP_SC, and WY_SC). For details, please refer to the study of Qiao et al. [16].

### 2.5. Quantifying the Key Features of Constraint Effect among Paired ESs

To analyze the dynamic changes of constraint relationships among ESs, thresholds were used to characterize the hump-shaped constraint relationship characteristics (Figure 4a), and k and b values on the constraint line were used to characterize the exponential type constraint relationship characteristics (Figure 4b). On the constraint line, when the y variable increases with the increase of *x* variable, it indicates that the constraint effect of *x* variable on *y* variable is decreasing; when the *y* variable decreases with the increase of *x* variable, it indicates that the constraint response of *x* variable on y variable is increasing. The k represents the strength of the constraint effect between paired ESs; when the k-value is greater than 0, the constraint effect decreases as the k-value increases, and vice versa, it increases as the k-value increases. The b-value characterizes the position of the constraint line and indicates the constraint effect when the ecosystem service of the *x*-axis is almost zero.

### 2.6. Identifying Key Drivers and Their Interactive Effects on Relationships between Paired ESs from 2000 to 2020

In this study, potential influencing factors include climatic factors (TEM and PPT), NDVI, landscape composition and configuration, and socio-economic factors (Table 2). Drawing on previous studies [4,20,42,43], this study selected the proportion of cropland, forest land, grassland, water area, urban land, and unused land area to reflect the regional landscape composition characteristics, and selected the Perimeter-area fractal dimension (PAFRAC), Landscape shape index (LSI), Contagion (CONTAG), Shannon’s diversity index (SHDI), and Patch density (PD) to reflect the landscape configuration characteristics. The details of the landscape index were shown in Appendix A and could be calculated by Fragstats 4.2 software, created by Kevin Mcgarigal and Eduard Ene from the United Sates.

With the help of the automatic linear modelling (ALM) model in SPSS 19.0 software, the influence of natural and socio-economic factors and their interaction on the characteristic values of the constraint relationships between ESs can be analyzed. The model can remove irrelevant variables while ensuring that there is no multicollinearity among the variables to obtain optimal regression results. In addition, a variable importance plot is generated at the end of the ALM model run, which indicates the contribution rate of the independent variable to the dependent variable and facilitates the elimination of non-significant variables. In this case, the total relative importance degree of each variable is 1 [42].

First, a basic model (Model 0) was created to analyze the effects of natural and socioeconomic factors on the changes in the key features of the constraint lines between pairs of ESs during 2000–2020. The model is as follows.
(10)$$Model\;0:\;\;\;\;\;\;\;Y=\alpha_{0}+\alpha_{1}X_{1}+\alpha_{2}X_{2}+\cdots +\alpha_{I}X_{I}+\delta ,$$
where *Y* represents the eigenvalue of the constraint line between pairs of ESs; $X_{i}$ represents the influence factor; $\beta_{i}$ is the coefficient of the model; $\beta_{0}$ is the intercept and δ is the error term.

Second, the model 0 was extended. The combination of various interaction effects between socioeconomic factors and landscape pattern indices were added to the model, and the new model was as follows.
(11)$$Model\;1:\;\;\;\;\;\;\;Y=\mu_{0}+\mu_{1}X_{1}+\mu_{2}X_{2}+\cdots +\mu_{m}X_{m}+\varepsilon +\mu_{j}X_{j}X_{0},$$
where $X_{j}X_{0}$ is the interaction between $X_{j}$ (socioeconomic factors) and $X_{0}$ (landscape pattern index); $\mu_{j}$ is the coefficient of the model; and ε is the error term.

## 3. Results

### 3.1. Spatiotemporal Patterns of ESs

From 2000 to 2020, the four ESs including NPP, FP, WY, and SC existed significant spatial and temporal differences in HHHP (Figure 5). NPP was high in the south and low in the north. In the northeast margin region, NPP was generally less than 2.00 t ha^−1^. In terms of temporal variation, NPP in the HHHP was the lowest in 2001 (3.59 t ha^−1^) and the highest in 2020 (4.85 t ha^−1^), showing a significant increasing trend (a = 0.03 *, * means significant at the 0.05 level). The low-value areas of FP were mainly distributed in Tianjin, Hebei, and Beijing in the north of the study area, while the high value areas were mainly concentrated in the regions of Anhui, and Henan. From 2000–2020, the annual average FP was 4.65 t ha^−1^, with the lowest occurring in 2002 (3.20 t ha^−1^) and the highest in 2020 (5.80 t ha^−1^), showing an overall significant increasing trend (a = 0.13 **, ** means significant at the 0.01 level). SC exhibited a distribution pattern of high in the central mountains and low in the surrounding plains. The SC was generally lower than 50.00 t ha^−1^ in the rest of the area, except for the central mountainous area of HHHP where the soil retention was higher. In 2002, the SC was as low as 5.31 t ha^−1^ and reached 28.96 t ha^−1^ in 2013, with large interannual fluctuations. The annual mean WY was 375.25 mm. Except for 2004 when WY was higher in the central region, WY showed a spatially increasing distribution pattern from north to south in all other years. The low WY area was mainly concentrated in the northern of HHHP, where the water yield was generally below 250 mm.

### 3.2. The Constraint Effect among Paired ESs from 2000 to 2020

The constraint line can accurately reflect the boundary of each pair of ESs scatter cloud, and the goodness of fit (R^2^) is high (Table 3). In this study, six pairs of ESs contain two types of constraint relationships. SC_FP presents an exponential curve type, and FP decreases with the increase of SC (Appendix A). The rest of the ESs show a hump-shaped constraint relationship type, which has a threshold effect, and the threshold changes over time (Figure 6, Appendix A). From 2000 to 2020, the constraining effect of increasing NPP on the remaining three ESs (FP, SC, and WY) first decreased and then increased, i.e., when NPP did not exceed the threshold, NPP gradually increased and its constraining effect on FP, SC and WY gradually decreased; when NPP exceeded the threshold, NPP gradually increased and its constraining effect on the remaining three ESs gradually increases. WY_FP and WY_SC are similar in the change of the constraint effect, both show that the constraint effect increases first and then decreases.

### 3.3. Key Features of the Constraint Lines among Paired ESs from 2000 to 2020

From the box plot of constraint line thresholds characterizing ES relationships, we found that the NPP thresholds on the NPP_SC, NPP_WY, and NPP_FP constraint lines changed less from 2000 to 2020, especially for NPP_SC. The NPP thresholds were concentrated in the range of 4.29–5.06 (Figure 7 and Figure 8 and Table 4). The NPP threshold on the NPP_WY constraint line showed a significant increase trend (*p* < 0.01). In comparison, the SC thresholds on the NPP_SC and WY_SC constraint lines varied widely, with the largest deviation from the SC threshold in the WY_SC constraint line. The WY thresholds on the NPP_WY, WY_FP, and WY_SC constraint lines were relatively stable, and the median of the SC thresholds on the WY_SC constraint line and the FP thresholds on the WY_FP constraint line was located at the bottom of the box. The median of the FP thresholds on the NPP_FP constraint line had the opposite trend. The FP thresholds on both WY_FP and NPP_FP constraint lines showed a significant increasing trend (*p* < 0.01). Among them, the FP thresholds of WY_FP and NPP_FP reached the minimum values in 2003 and 2002 with 5.35 t ha^−1^ and 5.54 t ha^−1^, respectively; and the maximum values in 2018 and 2019 with 9.49 t ha^−1^ and 9.30 t ha^−1^, respectively. SC_FP features were represented by k and b values. From 2000 to 2020, the k value of SC_FP was greater than 0 and tends to increase, and the constraint relationship between SC and FP gradually decreased. The b-value of the SC_FP tended to increase significantly, indicating that the maximum value that FP could reach in the absence of the limiting factor is increasing.

### 3.4. Effects of Driving Factors on the Constraint Relationship among Paired ESs

Natural factors and human activities were important factors that influence the constraint relationships between ESs. From 2000–2020, the proportion of water area was the main factor influencing the change of the NPP threshold on the NPP_FP constraint line with a contribution of up to 26.3%, followed by PPT and LSI. For the FP threshold of NPP_FP, the proportion of water area (36.3%) was more influential, followed by the proportion of forest land area (26.0%), PAFRAC (22.7%), the proportion of the unused land area (7.5%), and GDP (5.8%). The NPP threshold on the NPP_SC constraint line was mainly negatively influenced by the TEM with a contribution of up to 98.7%. For the SC threshold in the NPP_SC constraint relationship, it was significantly influenced by NDVI and PPT only, with contributions of 69.6% and 30.4%, respectively. The magnitude of the k-value of the SC_FP constraint line was mainly influenced by the proportion of the unused land area. This was followed by socioeconomic factors and population, while the remaining factors had weak or no effect on the magnitude of the k value of the constraint line. The proportion of water area and PAFRAC had positive effects on the b value of the SC_FP constraint line with a contribution of 34.7% and 16.7%, respectively. In contrast, the proportion of forest land area and GDP was negatively affected, with contributions of 25.6% and 14.9%, respectively. The main influences of both the NPP threshold and WY threshold on the NPP_WY constraint line were PPT. Among the factors affecting the WY threshold, population, LSI, and CONTAG played a negative role, while the remaining factors showed a positive effect. WY threshold on the WY_FP constraint line, the proportion of cropland area and NDVI contributed more to it with 79.9% and 10.1%, respectively, and the proportion of cropland area had a negative effect and NDVI had a positive effect.

PPT was a key factor in determining the FP threshold on the WY_FP constraint line, with a contribution of up to 97.5%. The WY threshold on the WY_SC constraint line was most influenced by the TEM, with a contribution of up to 41.3%, and the rest of the factors had insignificant effects on it. The socioeconomic factors of the population (3.0) and GDP (−4.3) both had some significant effects. For the SC threshold on the WY_SC constraint line, it was only significantly influenced by NDVI and population (Table 5).

### 3.5. Interaction Effects of Socioeconomics and Landscape Configuration

Among the six pairs of ESs constraint relations, the interaction between socioeconomic factors and landscape pattern indexes only had a significant effect on the change of the WY threshold for NPP_WY and WY_SC (Model 1). The effect of the interaction combination of population and PAFRAC on the WY threshold of WY_SC was stronger than that of GDP and PAFRAC on WY threshold of NPP_WY. For the WY threshold on the WY_SC constraint line, the interaction between population and PAFRAC had a significant negative effect on the WY threshold, and the more population, the more significant the influence of PAFRAC. When GDP increased, PAFRAC exerted a greater significant positive effect on the WY threshold in the NPP_WY constraint line (Table 5).

## 4. Discussion

### 4.1. Mechanisms of Constraint Relationship among ESs

Research on relationships between ESs is a frontier field of ecological research, and many scholars have carried out a lot of studies on the synergistic/tradeoff relationship between Ess [10,44]. However, there are also non-linear relationships among ESs, which will change dynamically over time [6]. Only by clarifying the influencing mechanism of dynamic changes of ESs, can regional ecosystem sustainability be improved. In this case, this study selected the HHHP as a typical area and attempted to identify the nonlinear relationship between ESs and evaluate the dynamic changes of the relationships, to reveal the mechanism of the impact of human activities and natural factors on the change of the relationships between ESs. The results indicated that in HHHP, the scatter points of the six pairs of ESs exhibited “scatter clouds”, indicating that the constraint line method could effectively reveal the interaction between ESs. The constraint relationships among NPP_SC, NPP_FP, and NPP_WY showed a hump-shaped curve type and had obvious constraint thresholds, which was consistent with the research results of Hao et al. [15]. In ecology, the threshold of constraint relationship represents the maximum value of response variables, around which the ecosystem shifts from one steady-state to another. The structure, functions, and services of an ecosystem differ greatly on either side of the threshold [45,46]. The thresholds of constraint relationship among ESs were crucial to optimize the supply of regional ESs and allocate resources rationally. On the left side of the threshold, an increase in NPP indicated that the vegetation cover was increasing and its constraint effect on FP, WY, and SC was decreasing. As to the reason, the HHHP was prone to drought risk and PPT was an important factor affecting the growth of vegetation [47]. An increase in NPP means an increase in regional PPT, which will improve regional FP and water capacity. Meanwhile, the increase in vegetation cover will inhibit the occurrence of soil erosion and improve soil and water conservation capacity. At the right side of the threshold, plant evapotranspiration increased with vegetation growth as NPP increased further. WY was equal to PPT minus vegetation and soil evapotranspiration, so WY would decrease. As for SC, when NPP exceeded a certain threshold, higher NPP implied more local PPT and increased the possibility of soil water erosion, which led to a weaker inhibitory effect of NPP on SC, as evidenced by the study of Hao et al. who also demonstrated our findings [15].

The WY_SC constraint relationship also showed a hump-shaped curve. It has been shown that the linear constraint effect of SC_WY of Inner Mongolia grassland is negative [48]. This difference may be attributed to the spatial scale dependence of the constraint effect between paired ESs. In this study, when the WY was to the left of the threshold, both the WY and NPP increased with the increase of PPT, and the better the vegetation growth condition, the better the soil retention capacity. When the WY exceeded the threshold, it meant that the PPT would further increase, and the increase of surface runoff led to the occurrence of soil erosion. There was also a threshold effect on the constraint relationship between WY_FP, where the constraint effect of WY on SC first decreased and then increased. On the left side of the threshold, an increase in WY meant an increase in PPT, which would promote the increase of NPP and improve soil and water conservation ability. When SC reached the threshold, with the further increase of WY, excessive PPT would easily lead to the occurrence of soil water erosion and reduced SC.

Compared with the other five pairs of ESs, the constraint relationship between SC_FP showed an exponential curve type, and the constraint effect of SC on FP became stronger as the SC increased. The reason is that the region with high SC is concentrated in the central region of the HHHP (Figure 5), which is mainly mountainous (Figure 1) with relatively steep terrain. Steep slope areas are characterized by thin soil surface layers, relatively low infiltration rates, and high runoff, which tend to result in a decrease in crop productivity [49], thus showing that the higher the SC, the lower the FP. Qiao et al. also proved that crop yield would decrease with the increase of terrain slope, and the higher the slope, the stronger the constraint on crop yield [16].

### 4.2. Effects of Influencing Factors on the Dynamic Change of Constraint Relationships among ESs

The relationships between ESs varied over time in the HHHP (Figure 8). To further explore the formation mechanisms of the changes in the constraint relationships among ESs, the climatic factors, vegetation cover factors, landscape composition and configuration, and socioeconomic factors were selected in this study to analyze their effects on the variation in the constraint relationships among ESs. The results indicated that although the constraint relationship type between ESs did not change from 2000 to 2020, the key features of constraint relationships appeared significant changes. Among climate factors, PPT can directly affect water input and surface hydrology, and almost all ESs constraint thresholds, increasing or decreasing the potential maximum value of ESs. which is similar to Comes et al. and Lang et al. [50,51]. For example, the hump-shaped constraint relationship between NPP_WY, and PPT (−2.4 **) was the key factor affecting the position of the NPP threshold on the *x*-axis (Table 5). Water availability in the HHHP is an important biophysical factor that determines vegetation growth, with PPT being the main source of water. PPT in turn determines WY in the InVEST model. Therefore, high PPT is often accompanied by the possibility of high NPP and high WY. Meanwhile, PPT (−1.0 **) determined the height of the hump-shaped constraint relationship between WY_FP. 70% of PPT in the HHHP is concentrated in the summer [47], and excessive PPT can cause flooding, thereby inhibiting FP while promoting WY. The HHHP is hot and prone to drought, and moisture is an important constraint for local vegetation growth [52]. When PPT increased, NPP and FP increased subsequently, which could have a decisive effect on the threshold of the constraint relationship between NPP_FP. In addition, both TEM and NDVI had a significant impact on the constraint line threshold. The rising TEM would drive evapotranspiration of regional surface and vegetation, further aggravated the occurrence of drought, which was not conducive to the growth of local vegetation, thus affecting ESs function. For NDVI, the areas with high NDVI were concentrated in the central mountainous region of the HHHP with high topographic relief. Previous studies have shown that NDVI is positively correlated with NPP [6]. High NDVI implies an increase in PPT, which will promote the possibility of erosion occurrence and thus exerts a negative impact on SC. On the contrary, increased PPT would promote the increase of the WY threshold. In terms of landscape composition, the area of cropland, forest land, and grassland in the HHHP continued to decrease from 2000 to 2020, and landscape type changes could be fed back to ESs through a series of ecological processes [53], which in turn had an impact on the constraint relationship eigenvalues. For example, the proportion of cropland area was the main factor that determined the WY threshold in the WY_FP constraint line, which was related to the need for water consumption for agricultural production. Therefore, by changing the composition of the regional landscape pattern, it was possible to alter the relationships between ESs and enhance the maximum value of specific ESs, achieving a win-win or multi-win effect. In addition, for the landscape configuration, the landscape shape complexity was favorable to increasing the potential maxima of FP and WY, which was consistent with the previous studies [42]. During 2000–2020, influenced by human activities, the landscape pattern of HHHP became more irregular. Landscape patterns would exert important effects on ecosystem composition, structure, and function, ultimately leading to changes in the inter-ESs role relationships [19,54,55].

The k and b values together indicated the shape and location of the exponential constraint line. GDP had a significant negative effect on the k-value of the SC_FP constraint relationship, and an increase in GDP enhanced the constraint effect of SC on FP. This may be attributed to the reason that increased GDP will promote the improvement of agricultural technology. More and more cropland irrigation adopted drip irrigation mode [56], which both improved the efficiency of water use and reduced the occurrence of soil erosion in the HHHP. In addition, tillage practices such as conservation tillage were also beneficial to the improvement of SC. Urbanization led to an increase in GDP while occupying a large area of cultivated land, resulting in a decline in grain output. For the starting position (b value) on the SC_FP constraint line, the proportion of forest area had a significant negative effect on it, and the increase in the proportion of forest area would enhance the constraint effect between SC_FP. This was due to the implementation of the project of returning farmland to forest and grass in some areas of the HHHP [57,58,59], which improved SC. Conversely, the decrease in arable land area would lead to the decline of FP.

### 4.3. Implications for Ecosystem Management

The direction of the ESs constraint relationship represented by the threshold point of the constraint line would change around the threshold [48]. Decision-makers should scientifically recognize and understand the relationship between ESs when optimizing specific ES. The NPP thresholds in NPP_FP, NPP_SC, and NPP_WY were a key reference to achieving a win-win situation for NPP, FP, SC, and WY, and should be considered in ecological management. When NPP exceeded the threshold, the remaining three ESs declined; therefore, decision-makers could apply the stable variation range of the NPP threshold to the overall optimization of ESs. As an important grain-producing region in China, the HHHP played a vital role in ensuring national food security. Agricultural policies should be formulated in such a way as to minimize the adverse effects of scattered and fragmented agricultural land and to develop intensive agricultural production. Meanwhile, cropland management measures should be improved by adopting drip irrigation, micro-irrigation, and conservation tillage measures to enhance the overall supply potential of regional ESs and achieve regional sustainable development. Due to the data availability, this study only selected the time range from 2000 to 2020 for analysis. However, the type of constraint relationship between ESs may change as the time scale becomes longer. Future studies will consider a longer time scale to explore whether the types of interaction relationships among ESs will mutate over time.

## 5. Conclusions

This study adopted the constraint line method to identify the constraint relationship types and characteristic values of ESs in the HHHP. Using the automatic linear regression modelling, we revealed the dynamic changes and influencing factors of ESs during 2000–2020. The results appeared that NPP, FP, and SC increased except for WY, which exhibited a non-significant decreasing trend in the past 20 years. There were two types of constrained relationships among the six pairs of ESs, humped-shaped and exponential. The threshold constraint effect existed among NPP, FP, SC, and WY. By adjusting NPP, the potential maximum value of specific ESs could be improved to achieve win-win or multi-win of ESs. From 2000 to 2020, although the type of constraint relationships between ESs did not change, the key features of the constraint relationships, namely threshold, slope (k), and constant term (b), occurred significant changes. In the HHHP, PPT is an important factor influencing the variation of constraint relationships, especially critical for FP potential. In most cases, PPT not only weakened the constraint effect between ESs, but also raised the starting position of the constraint line. For the landscape configuration, the landscape shape complexity was beneficial to increasing the potential of FP and WY. Overall, understanding the impacts of human activities and natural factors on the relationships between ESs could provide a basis for the formulation of ecological management strategies and the coordinated development of ESs in the HHHP, which was conducive to ensuring national food security and improving regional ecosystem sustainability.

## Figures and Tables

**Figure 1 ijerph-19-10230-f001:**
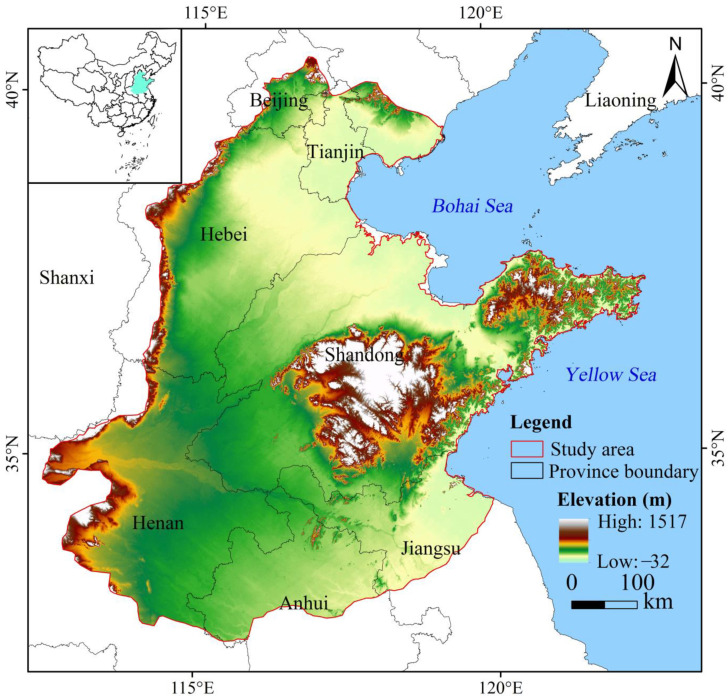
Location of the study area.

**Figure 2 ijerph-19-10230-f002:**
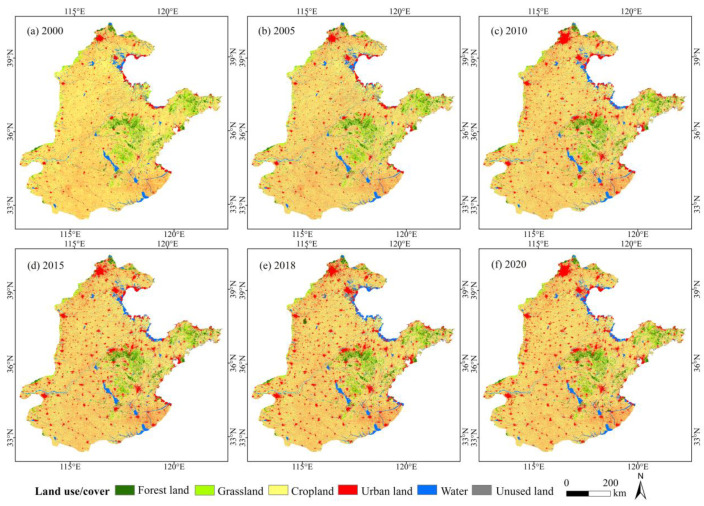
Land use/cover in HHHP in 2000 (**a**), 2005 (**b**), 2010 (**c**), 2015 (**d**), 2018 (**e**) and 2020 (**f**).

**Figure 3 ijerph-19-10230-f003:**
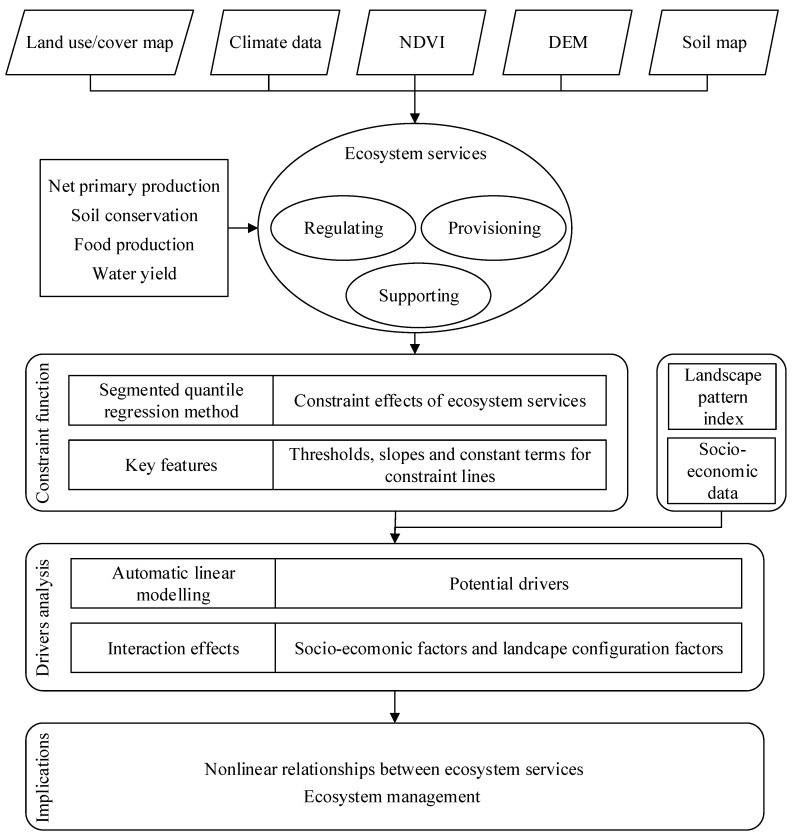
Methodological framework.

**Figure 4 ijerph-19-10230-f004:**
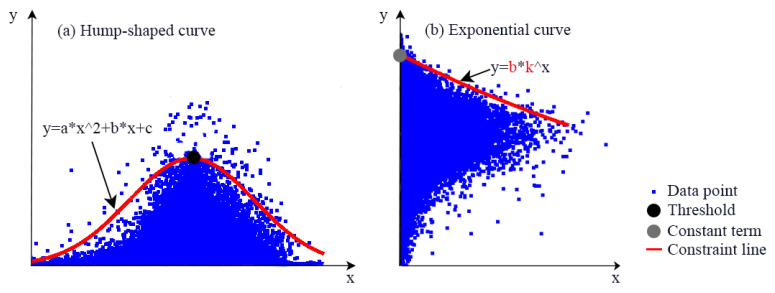
Extraction of constraint lines using the segmented quantile regression approach.

**Figure 5 ijerph-19-10230-f005:**
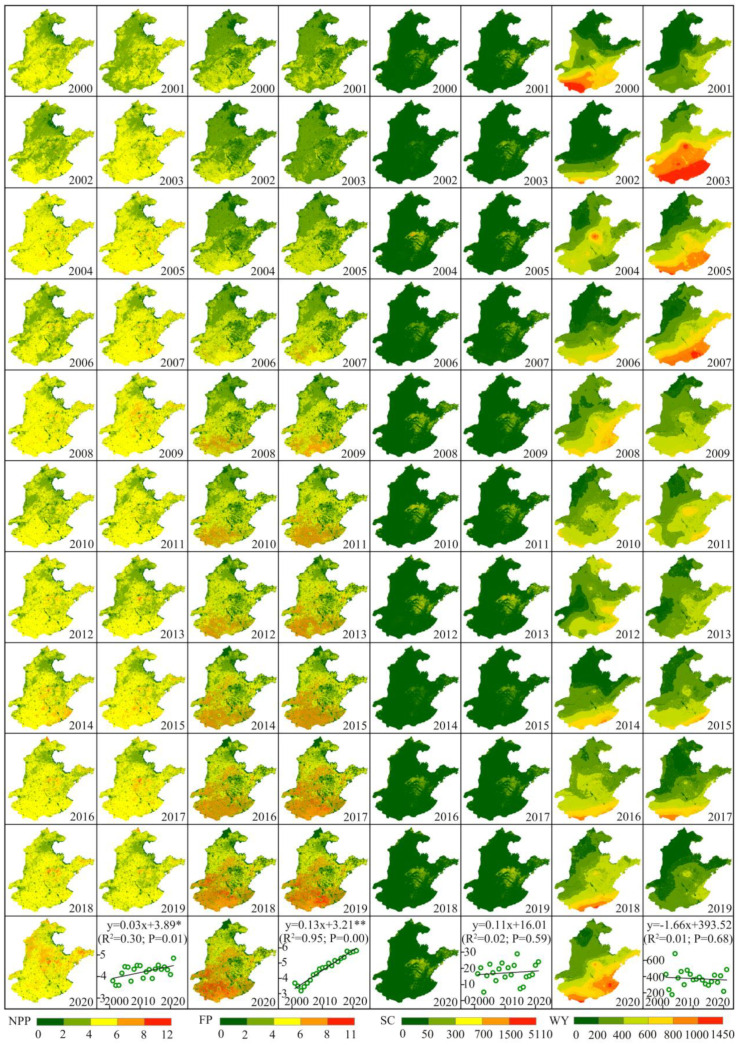
Spatial distribution of ESs from 2000 to 2020. NPP: net primary productivity; FP: food production; SC: soil conservation; WY: water yield. The units of NPP, FP, and SC are t ha^−1^, and the unit of WY is mm. ** means significant at the 0.01 level; * means significant at the 0.05 level.

**Figure 6 ijerph-19-10230-f006:**
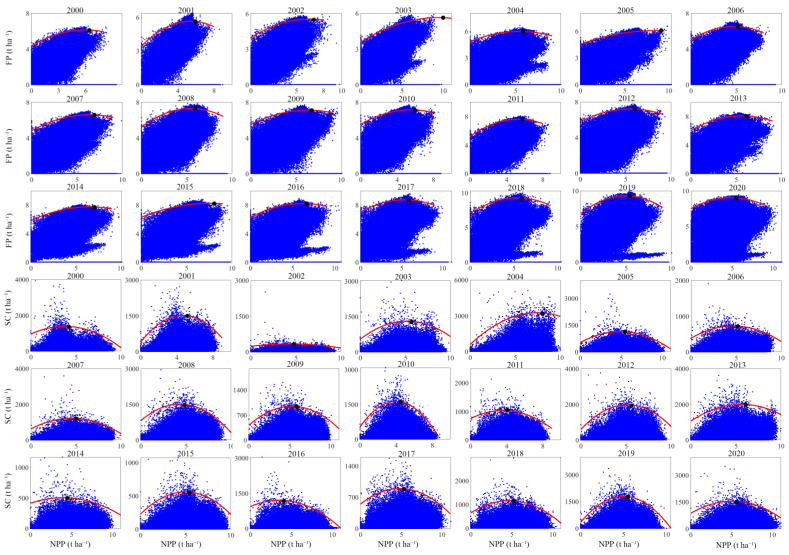
The scatter clouds (blue points), thresholds (black points), and constraint lines (red lines) between paired ESs (A_B) from 2000 to 2020. A indicates the constraint ES on the *x*-axis and B indicates the corresponding ES on the *y*-axis. NPP: net primary productivity; FP: food production; SC: soil conservation. The units of NPP, SC and FP are t ha^−1^ year^−1^.

**Figure 7 ijerph-19-10230-f007:**
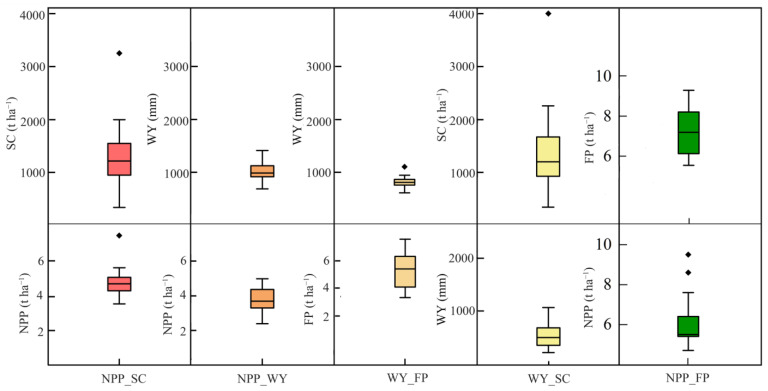
The box plots of thresholds on the constraint lines of NPP_SC, NPP_WY, WY_FP, NPP_FP and WY_SC. NPP: net primary productivity; FP: food production; SC: soil conservation; WY: water yield. Diamond represents outliner.

**Figure 8 ijerph-19-10230-f008:**
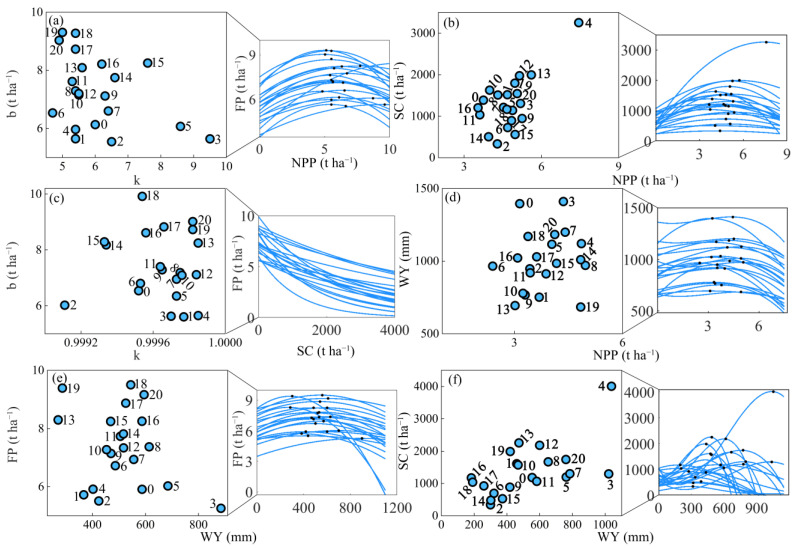
The constraint lines (**a**–**f**) and scatter plots of key features between paired ESs (X_Y: X represents the ES on *x*-axis and Y represents the ES on *y*-axis) and their thresholds (black points) from 2000 to 2020. NPP: net primary productivity; FP: food production; SC: soil conservation; WY: water yield. The units of SC, FP and NPP are t ha^−1^, and the unit of WY is mm. The numbers 0–20 represent the years 2000–2020, respectively.

**Table 1 ijerph-19-10230-t001:** Description of the data used in this study.

Data	Data Description (Unit)	Data Source
Meteorological data	Daily mean temperature (°C)	China Meteorological Sharing Service System
	Daily rainfall (mm)	
	Daily sunshine duration (h)	
Digital Elevation Model (DEM)	The Shuttle Radar Topography Mission (SRTM) digital elevation model with 90-m spatial resolution (m)	Geospatial Data Cloud (https://www.gscloud.cn/home, accessed on 1 October 2020)
Soil data	HWSD (v1.1) soil dataset (including fractions of sand, silt, clay and organic carbon in the topsoil and soil depth)	Cold and Arid Regions Science Data Center at Lanzhou
Land use/cover	Land use/cover in 2000, 2005, 2010, 2015, and 2018 at 30-m spatial resolution, and 2020 at 250-m spatial resolution	Resource and Environment Science and Data Center
Normalized Difference Vegetation Index (NDVI)	MODIS NDVI product (250 mm resolution MOD13Q1 product)	NASA
Crop yield	The crop yield of staple food crops for each city	Statistical yearbook
Socio economic data	Gross Domestic Product (GDP)_	Resource and Environment Science and Data Center (https://www.resdc.cn/, accessed on 2 December 2021)
	Population	World Pop (https://www.worldpop.org/, accessed on 2 December 2021)

**Table 2 ijerph-19-10230-t002:** Description of potential driving factors investigated in this study.

Data	Data Description
Climatic factors	Average precipitation (mm)
	Average temperature (°C)
Vegetation factor	Normalized Difference Vegetation Index
Landscape composition	The total area of cropland (%)
	The total area of forest land (%)
	The total area of grassland (%)
	The total area of water (%)
	The total area of urban land (%)
	The total area of unused land (%)
Landscape configuration	Perimeter-Area Fractal Dimension
	Landscape Shape Index
	Contagion (%)
	Shannon’s Diversity Index
	Patch Density (Unit/100 ha)
Socio-economic factors	GDP (CYN)
	Population (person)

**Table 3 ijerph-19-10230-t003:** The goodness of fit values (R^2^) of the constraint lines.

	NPP_FP	NPP_SC	SC_FP	NPP_WY	WY_FP	WY_SC
2000	0.79 **	0.21 **	0.77 **	0.94 **	0.75 **	0.41 **
2001	0.80 **	0.56 **	0.73 **	0.88 **	0.43 **	0.61 **
2002	0.84 **	0.13 **	0.52 **	0.91 **	0.41 **	0.40 **
2003	0.90 **	0.22 **	0.72 **	0.93 **	0.73 **	0.56 **
2004	0.63 **	0.38 **	0.84 **	0.51 **	0.82 **	0.91 **
2005	0.86 **	0.38 **	0.78 **	0.95 **	0.79 **	0.44 **
2006	0.71 **	0.30 **	0.70 **	0.95 **	0.72 **	0.75 **
2007	0.81 **	0.31 **	0.66 **	0.96 **	0.42 **	0.66 **
2008	0.76 **	0.49 **	0.80 **	0.89 **	0.77 **	0.68 **
2009	0.85 **	0.38 **	0.77 **	0.93 **	0.63 **	0.66 **
2010	0.76 **	0.75 **	0.89 **	0.91 **	0.67 **	0.66 **
2011	0.87 **	0.28 **	0.87 **	0.89 **	0.79 **	0.58 **
2012	0.78 **	0.38 **	0.76 **	0.89 **	0.67 **	0.80 **
2013	0.82 **	0.33 **	0.66 **	0.80 **	0.63 **	0.67 **
2014	0.81 **	0.17 **	0.62 **	0.89 **	0.59 **	0.54 **
2015	0.82 **	0.47 **	0.83 **	0.92 **	0.74 **	0.57 **
2016	0.80 **	0.50 **	0.76 **	0.95 **	0.74 **	0.59 **
2017	0.64 **	0.42 **	0.78 **	0.94 **	0.64 **	0.77 **
2018	0.77 **	0.44 **	0.67 **	0.96 **	0.42 **	0.62 **
2019	0.79 **	0.67 **	0.71 **	0.89 **	0.66 **	0.77 **
2020	0.82 **	0.30 **	0.76 **	0.97 **	0.58 **	0.56 **

Notes: NPP: net primary productivity; FP: food production; SC: soil conservation; WY: water yield. ** means significant at the 0.01 level.

**Table 4 ijerph-19-10230-t004:** The thresholds, slopes (k) and constant terms (b) of FP_SC, FP_WY, WY_SC, NPP_WY and NPP_SC.

	Thresholds										Slopes (k) and Constant Terms (b)	
	NPP_SC		NPP_WY		WY_SC		WY_FP		NPP_FP		SC_FP	
	NPP	SC	FP	WY	WY	SC	WY	FP	NPP	FP	k	b
2000	3.75	1386.35	3.54	1379.62	554.12	1183.14	587.20	5.91	6.00	6.13	0.9995	6.53
2000	4.68	1517.80	4.12	803.00	460.27	1611.79	367.30	5.72	5.40	5.64	0.9997	5.60
2002	4.29	332.45	3.96	978.35	300.01	338.98	424.20	5.50	6.50	5.54	0.9991	6.02
2003	5.19	1306.24	3.49	1439.64	1021.36	1296.03	884.70	5.25	9.50	5.64	0.9997	5.62
2004	7.45	3254.79	2.88	1100.41	1038.03	4006.01	402.50	5.91	5.40	5.96	0.9998	5.64
2005	4.88	1139.83	4.04	1146.43	761.87	1197.49	684.80	6.02	8.60	6.06	0.9997	6.34
2006	4.69	725.93	4.06	954.79	321.33	695.95	486.40	6.72	4.70	6.53	0.9995	6.80
2007	4.53	1212.90	4.00	1218.52	784.28	1301.24	555.80	6.93	6.40	6.59	0.9997	6.95
2008	4.31	1510.45	4.56	962.81	652.23	1669.39	614.10	7.37	5.40	7.29	0.9997	7.18
2009	5.25	945.13	4.61	772.81	418.19	884.83	470.10	7.14	6.30	7.11	0.9996	7.28
2010	3.99	1627.28	4.77	781.59	468.34	1572.66	453.30	7.27	5.50	7.15	0.9997	7.09
2011	3.61	1033.68	4.75	927.59	582.29	1066.09	505.50	7.73	5.30	7.61	0.9996	7.39
2012	5.15	1975.39	4.68	916.55	599.66	2183.67	517.50	7.33	5.50	7.19	0.9998	7.11
2013	5.61	1996.45	4.88	696.88	473.73	2257.91	270.80	8.29	5.60	8.09	0.9998	8.24
2014	3.95	504.21	4.57	1025.30	301.90	478.84	517.10	7.81	6.60	7.74	0.9993	8.17
2015	4.98	559.82	4.69	1006.75	373.17	527.14	468.90	8.24	7.60	8.25	0.9993	8.28
2016	3.54	1202.79	4.76	1042.39	183.32	1173.02	587.00	8.25	6.20	8.21	0.9995	8.61
2017	4.84	893.33	5.73	1042.21	259.75	924.20	525.70	8.86	5.40	8.72	0.9996	8.82
2018	4.66	1165.07	6.08	1200.92	191.20	1039.53	544.70	9.49	5.40	9.27	0.9995	9.91
2019	4.97	1795.86	5.98	693.92	420.26	1989.14	286.70	9.38	5.00	9.30	0.9998	8.72
2020	5.06	1545.14	6.07	1194.82	760.21	1742.78	594.80	9.16	4.90	9.03	0.9998	9.01
Trend	↓	↓	↑ **	↓	↓	↓	↓	↑ **	↓	↑ **	↑	↑ **

Note: the units of NPP, FP and SC are t ha-1, and the unit of WY is mm. In the line of trend, “↓” indicates the falling trend; “↑” indicates the rising trend. NPP: net primary productivity; FP: food production; SC: soil conservation; WY: water yield. ** means significant at the 0.01 level.

**Table 5 ijerph-19-10230-t005:** Results of automatic linear modelling models.

	Thresholds	k	b	
	NPP_FP	NPP_SC	WY_SC	NPP_WY	WY_FP	SC_FP	
	NPP	FP	NPP	SC	WY	SC	NPP	WY	WY	FP			
Model	0	0	0	0	0	1	0	0	0	1	0	0	0	0	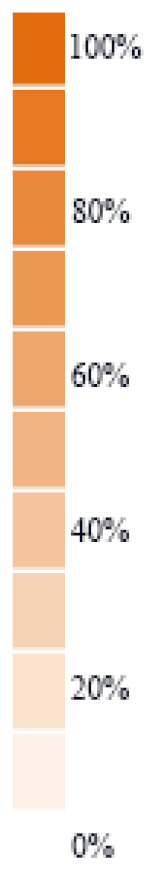
Accuracy (%)	40.9	95.7	99.9	100.0	91.9	91.9	100.0	94.2	82.4	82.4	92.0	70.0	39.1	94.2
NDVI		0.12		−2.1 **	0.3	0.3	−3.7 **	1.8 **			0.4 *			
PPT	0.7 *		−0.2	1.4 *	0.2	0.2		−2.4 **	1.0 **	1.0 **		0.9 **		0.1
TEM	−0.4		−1.6 *		−0.4 **	−0.4 **			0.8 *	0.8 *		−0.1		
population	0.6				−4.3 *		6.2 **		−6.8 *	−6.8 *			3.7	−1.3
GDP		0.3*			3.0 *	3.0 *			7.6 *				−3.5 *	1.6 *
Area_(cropland)_											−1.0 **			
Area_(water)_	−1.4 *	0.8 **												0.8 **
Area_(forest)_		−1.4 **											−0.6	−1.2 **
Area_(unusedland)_	−1.4	0.3 *											1.1 *	
CONTAG									−8.8 *	−8.8 *				
LSI	1.9 *								−13.5 *	−13.5 *				
PARFAC	−0.8	1.0 **			−4.0	−3.5			8.6 *	8.6 *				0.9 **
Interaction effects														
Population *PAFRAC						−4.4 *								
GDP*PAFRAC										1.3 *				

Notes: Color refers to the contribution rate of key features on the constraint line (%). NPP: net primary productivity; FP: food production; SC: soil conservation; WY: water yield. ** means significant at the 0.01 level; * means significant at the 0.05 level. Model 0 was constructed to evaluate the response of key features on constraint lines to climatic factors, vegetation factor, landscape composition, landscape configuration and socio-economic factors. The interaction between landscape indicators and socio-economic factors was added into Model 0 to form model 1.

## Data Availability

Not applicable.

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
