# Peer review of "Revealing Impacts of Human Activities and Natural Factors on Dynamic Changes of Relationships among Ecosystem Services: A Case Study in the Huang-Huai-Hai Plain, China"

_ijerph, 2022, doi:10.3390/ijerph191610230_

Round 1

Reviewer 1 Report

GENERAL COMMENTS

The manuscript analyzed characteristics of the constraint relationships among six pairs of ESs during 2000-2020 and further revealed the impacts of human activities and natural factors as well as their interactions on the constraint relationships among ESs.  Overall, the topic is interesting and the manuscript has a well-organized structure.  I have a few suggestions.

- My first concern is that the methods part lacks details and should provide more specific information for different parameters of NPP, SC, and WY.  For example, “t” is a specific month in the calculation of NPP, and which month does “t” consider?  How do the authors determine the values of R, K, LS, C, and P for SC and those of Kcx, ETox, AWCx, and Z for WY?  More importantly, what does the constraint effect represent?  The manuscript mentioned that the constraining effect of increasing NPP on the three ESs first decreased and then increased, i.e., when NPP did not exceed the threshold, NPP gradually increased and its constraining effect on the three ESs gradually decreased.  Does it mean that the constraint effect of one ES on the other ES will decrease if there is a synergistic relationship and increase if there is a trade-off relationship between them?  How to estimate the strength of constraint effects?  What do the slope and k represent?  I highly recommend the authors give more detailed information on the constraint effect and ESs as well as their calculations.

- My second concern is that there are some grammatical errors and ill-structured sentences throughout the entire manuscript.  For example, the title should be “Revealing IMPACTS of human activities and natural factors on DYNAMIC changes of relationships among ecosystem services”.  I recommend the authors have a native English-speaking researcher review the manuscript or use a professional English editing service to improve their English.

SPECIFIC COMMENTS

- Lines 148-149: Please add all data used in this study (e.g., population and GDP data with the spatial resolution) in Table 1.  The MODIS NDVI version used should also be provided.  Why not use the land use/cover data in 2020 at 30-m spatial resolution?  I think data acquisition can be added as a single part (e.g., 2.2. Data acquisition).

- Lines 231-249: Why use the automatic linear modeling (ALM) model?  What’s the difference between the ALM and general multiple stepwise linear regression?  Socioeconomic factors (Xj) contain GDP and population, and landscape pattern indices (Xo) include different metrics.  How to estimate the two indicators of Xj and Xo?

- Line 271: Please add R2 and P-value to the three regression lines in Fig. 5.

- Line 296: Please add P-value for each R2 in Table 3.

- Lines 321-325: Please explain what numbers represent in each plot.

- Line 334: How to estimate the contribution rate?  Please add the explanation in the 2.5 part.

- Lines 442-443: The annual average precipitation in the study area is about 500-1000mm.  Why is the HHHP prone to drought?

- Lines 475-481: Why does urbanization result in a decline in grain output or does the decrease in the arable land area lead to the decline of FP?  With the increasing demand of the population for food and technology improvement, the FP tends to increase during the past several decades.

- Lines 515-516: Why does PPT weaken the constraint effect between ESs and lower the starting position of the constraint line?  In Table 5, most coefficients of PPT for thresholds of constraint effects of NPP_FP were larger than 0.  What do coefficients with larger or less than 0 represent?

Reviewer 2 Report

The paper presents an interesting overview of ecosystem services and the impacts on landscape changes caused by human activities, using the latest remote sensing methods, statistics and current online sources, using the example of a province in China that has developed dynamically over the past 20 years. The paper shows a clear goals and raised research questions. The advantage of the manuscript is a transparent research framework that has been thoroughly tested in the paper with many extensive research methods, statistical analyses, modern tools and sources in the field of remote sensing. The given manuscript is written in an understandable and clear language, the layout of the paper is appropriate and legible, and the used methods and statistical tools are also well-illustrated. The paper is represented by numerous figures and tables that complement the content of the research. The methods and their discussion are well presented in the paper and discussed in the selected case study. The results and conclusions are also transparent and are complemented by summarised tables and coloured diagrams. The goal set of the study were achieved in the conducted research and summerized in discussion and conlusions.

Minor comments:

- In Chapter 1 (Introduction), for the sake of clarity of the paper, I would like to add two sub-chapters for the subdivision of the sections: (1.1) Purpose of the study and research questions, (1.2) General (spatial) characteristics of the research area ;

- I suggest to add in which GIS Software (name/version) the given maps (e.g. Figures 1, 2 and 5) are created, e.g. in Chapter 2 (Methods), and/or in given fugure, or directly in the description of the figures, and whether it is, for example, an own author’s elaboration based on geodata (which?);

- Many abbreviations of the names of selected indicators, which are important for the paper are appeared. However, their explanations are mailnly shown in the beginng of the manuscript (e.g. in Chapter 2, Methods). Note: Explanations of some of the most important abbreviations may be also included in tables and diagrams (or by adding the legend to the given figure description) to make the manuscript clearer;

- I do not find figures S1 and S2 in the manuscript (quotation in verses 279 and 281). I guess it is about: Table S1, Table S2 (?) Please correct.

Other suggestions:

- I think it is interesting to compare the given study area with another similiar province (e.g. in China), to broaden the view of research problem. However, due to well-presented content value of the study, this idea might be looked at separately study in the future.

Summing up, after minor corrections, I think the paper meets the editorial requirements and is suitable for publication in the IJERPH MDPI journal.

Reviewer 3 Report

Dear Authors, 

After reading the paper, I have comments and suggestions for improving the paper, which are as follows: 

The abstract is too long it should be revised according to the guidelines of the journal, as it lacks information about the research methods used and the results obtained.  

In the Introduction   

I suggest completing the following: add the research questions and the research hypothesis. 

The chapter Theoretical Background or Literature Review is missing.

Suggest adding this chapter. Present previous research on this topic in the world.

Materials and Methods 

These are well presented and described. I suggest introducing a scheme of research procedure.   

Results

The results are presented and described in a good way, they are very interesting and important for determining the impact of human activities and natural factors on the dynamic changes in the relationship between ecosystem services.

In the Discussion section, the authors should more thoroughly discuss and explain the conclusions and results of the work. This will improve the quality of the work. The authors should compare their project and results with the results of similar studies conducted on this topic in other parts of Asia and the world.   

Technical errors to be removed: 

[302] bad notation -interpunctuation 

[302, 303, 3014, 315] multiple references to Table 4- unnecessary [440, 442,446, 457, 479]- in one section 4.2 multiple references to the same Table 5- unnecessary 

[229] incorrect notation of Table S1

[279]- incorrect notation Figure S1

[207, 209] - errors in notations (Figure 4 (b))

Correct literature according to the rules of the journal 

In conclusion, I recommend the paper for publication in the journal Sustainable Development with minor changes. 

Kind regards, 

Reviewer
